# Super Deep Contrastive Information Bottleneck for Multi-modal Clustering

Zhengzheng Lou [1]   Ke Zhang [1]   Yucong Wu [1]   Shizhe Hu [1]

## Abstract

In an era of increasingly diverse information sources, multi-modal clustering (MMC) has become a key technology for processing multi-modal data. It can apply and integrate the feature information and potential relationships of different modalities. Although there is a wealth of research on MMC, due to the complexity of datasets, a major challenge remains in how to deeply explore the complex latent information and interdependencies between modalities. To address this issue, this paper proposes a method called super deep contrastive information bottleneck (SDCIB) for MMC, which aims to explore and utilize all types of latent information to the fullest extent. Specifically, the proposed SDCIB explicitly introduces the rich information contained in the encoder's hidden layers into the loss function for the first time, thoroughly mining both modal features and the hidden relationships between modalities. Moreover, the proposed SDCIB performs dual optimization by simultaneously considering consistency information from both the feature distribution and clustering assignment perspectives, the proposed SDCIB significantly improves clustering accuracy and robustness. We conducted experiments on 4 multi-modal datasets and the accuracy of the method on the ESP dataset improved by 9.3%. The results demonstrate the superiority and clever design of the proposed SDCIB. The source code is available on https://github.com/ShizheHu.

## 1. Introduction

In recent years, with the rapid advancement of information acquisition technology, data storage capacity has also developed quickly. The sources of information have gradually diversified, evolving from traditional single-modal data to multi-modal data, while the scale and complexity of the data have significantly increased (Srivastava & Salakhutdinov, 2012; Baltrusaitis et al., 2019; Ramesh et al., 2021). "Modality" refers to different types of data obtained from various perspectives of information acquisition and processing (Hacquard, 2010). For example, a video can capture data information from both the sound and image perspectives. However, traditional single-modal data analysis methods often struggle to comprehensively handle multi-modal data. Multi-modal clustering (MMC), as a key technology, can fully mine and integrate information from different modalities, revealing the potential complementarity between them. It has been widely applied in areas such as cross-modal retrieval (Hu et al., 2019; Chun et al., 2021; Yuan et al., 2022), intelligent recommendation (Liu et al., 2021), and biomedical science (Acosta et al., 2022; Si et al., 2023). Therefore, research on MMC methods provides vast prospects for practical applications.

Existing traditional MMC methods mainly focus on three categories: subspace learning, graphical models and matrix decomposition (Cai et al., 2011; Xia et al., 2023). However, as multi-modal datasets become increasingly complex, traditional MMC methods are prone to the "curse of dimensionality" in high-dimensional spaces and struggle to handle high-dimensional data effectively.

At the same time, with the continuous development of deep learning, deep MMC methods fully integrate the advantages of deep neural networks, providing more solutions for MMC (Ngiam et al., 2011; Andrew et al., 2013; Xie et al., 2016; Liu et al., 2023; Gao et al., 2020). For example, Wang et al. (Wang et al., 2021) combined the self-supervised t-SNE module with the self-expression layer to learn a shared low-dimensional representation. Mao et al. (Mao et al., 2021) adopted the idea of contrastive learning to maximize the shared information between modalities and minimize the redundancy within each modality, achieving efficient clustering by integrating a multi-modal shared encoder with variational optimization. In recent years, Rong et al. (Rong et al., 2022) utilized a variational autoencoder architecture based on the autoencoder and incorporated an attention mechanism to extract cluster-friendly representations from multi-omics data. However, the aforementioned methods fail to deeply understand the complex relationships within data samples across modalities, neglecting the close connec-

[1]School of Computer and Artificial Intelligence, Zhengzhou University, Zhengzhou, China. Correspondence to: Shizhe Hu < ieshizhehu@gmail.com, https://shizhehu.github.io/>.

tions between the data, and may even disrupt the internal associations.

Therefore, although a large number of MMC methods have emerged in both traditional and deep learning fields in recent years, there are still some limitations in the actual clustering process. On the one hand, most existing MMC methods simply achieve a consensus between different modalities to fuse the data. While this improves clustering performance to some extent, it fails to deeply explore and capture the complex latent information and interdependencies between modalities in multi-modal data. On the other hand, many deep multi-modal methods often only focus on single data information such as clustering or features, without considering the two simultaneously. As a result, much of the inherent latent structural information in the data samples is ignored or even disrupted, failing to capture the more important heterogeneity between modalities. In this study, we propose a method called super deep contrastive information bottleneck for MMC (SDCIB) to address the aforementioned issues. The proposed SDCIB efficiently mines the latent information between modalities through the hidden layers. Meanwhile, it simultaneously considers both feature distribution and clustering assignment to better capture the inherent structure of the data. Our main approach is designed for multi-modal data, where we first combine the information bottleneck theory to design a variational IB encoder, then incorporate the concept of contrastive to optimize the computation of consistency information. And finally obtain the final clustering results through a clustering layer. The proposed SDCIB is the first to explicitly introduce the hidden layer information from the encoder into the clustering task. On one hand, the proposed SDCIB fully incorporates the concept of "compression" from the information bottleneck theory to eliminate redundant information in the modalities. On the other hand, the proposed SDCIB simultaneously focuses on consistency information at both the feature and clustering levels, deeply connecting the hidden layer's latent information to uncover the data's potential feature information, thereby better assisting clustering. Finally, The proposed SDCIB designs a clustering layer to obtain accurate clustering results. The main contributions of this work are as follows:

- For the MMC problem, we proposed SDCIB, which is the first to explicitly introduce the information contained in the encoder's hidden layers into the loss function, fully exploring the implicit information in the modal features.

- The proposed SDCIB performs dual optimization by simultaneously considering consistency information from both the feature distribution and clustering assignment perspectives. This approach can better capture the latent information between modalities, significantly improving clustering accuracy and robustness.

- In the specific experiments, we selected several classic multi-modal datasets and compared the proposed SDCIB with the state-of-the-art traditional and deep MMC methods. The proposed SDCIB consistently demonstrated significant advantages in clustering performance.

## 2. Method

### 2.1. Revisit: Information Bottleneck

The information bottleneck (IB) principle (Tishby et al., 1999) is based on information theory and is widely used in various real-world applications, such as image clustering (Hu et al., 2021) and classification tasks (Lou et al., 2013). A detailed discussion of the IB principle can be found in our survey (Hu et al., 2024). Its core concept is to control the balance between the amount of information compression and relevance. Suppose there is a source variable $X$ and a label $Y$. The goal of the information bottleneck is to find a compressed representation $T$ through the source variable $X$, which compresses the source variable $X$ as much as possible while retaining the relevance between the compressed representation $T$ and the label $Y$ as much as possible. Therefore, the objective function of the IB theory can be formulated as:

$$\mathcal{L}_{min} = I(T; X) - \beta I(T; Y). \tag{1}$$

where $I(T; X)$ represents the mutual information between the compressed representation $T$ and the source variable $X$. The smaller it is, the greater the compression. $I(T; Y)$ represents the mutual information between the compressed representation $T$ and the label $Y$. The larger it is, the stronger the retained correlation. $\beta$ is used to control the balance between compression and retention.

In recent years, IB theory has been widely used in various MMC tasks (Federici et al., 2020; Yan et al., 2024; 2025). For example, Federici et al. (Federici et al., 2020) proposed a multi-modal IB method that can identify non-shared information between two modalities, but it only explores the correlation of different modalities through feature distribution, ignoring the consistency of cluster assignment, making the learned feature representation unfriendly to downstream clustering tasks. Yan et al. (Yan et al., 2024) proposed a multi-modal IB method that uses shared representations of multiple modalities to eliminate private information of a single modality. Yan et al. (Yan et al., 2025) further proposed an incremental IB method that builds acknowledge base to solve the clustering problem of incremental modalities. Both of the above works considered the consistency of feature distribution and cluster assignment at the same time, but they failed to consider the correlation between feature

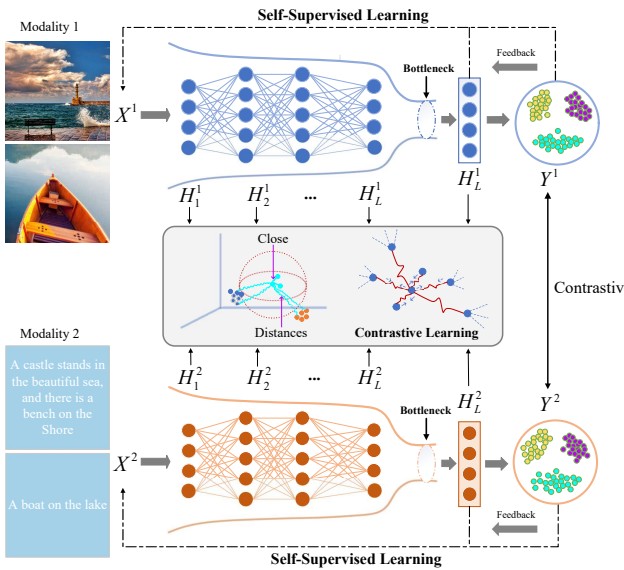

*Figure 1.* Framework of the proposed SDCIB. Given the original features $X^i$ of different modalities, they are input into the variational IB encoder to obtain the hidden layer features $H_1^i, H_2^i, \ldots,$ $H_{l-1}^i$ and the compressed representation $H_L^i$. $H_L^i$ passes through the clustering layer to obtain the clustering result $Y^i$, which in turn guides the encoder to compress $X^i$. In order to make different modalities as consistent as possible, we compare them from the perspectives of feature distribution and cluster allocation.

distribution and clustering results. All the above MMC IB methods ignore the rich information contained in the hidden layers of the encoder and fail to explicitly utilize it. The above limitations motivate us to propose SDCIB.

### 2.2. Problem Definition

For a given MMC task, suppose there are $M$ modalities and $N$ data samples to be clustered, which can eventually be clustered into $K$ clusters. Our goal is to learn better feature representations $H_L^i$ in $M$ modalities through the proposed SDCIB, to obtain good clustering results $Y^i$ for each modality, and finally combine the clustering results $Y^i$ for each modality to obtain the final clustering result $Y$ by:

$$y_i = \underset{j \in \{1,2,...,K\}}{\arg\max} \sum_{m=1}^{M} \mathbb{I}\left(y_i^m = j\right) \qquad (2)$$

where $y_i$ indicates which cluster $i$-th sample belongs to in the final clustering partition, and $y_i^m$ indicates which cluster $i$-th sample belongs to in the clustering partition of $m$-th modal.

### 2.3. Network Architecture

Our network framework is shown in Figure 1, which can be divided into three parts. 1) Variational IB encoder: The original features $X^i$ of the $i$th modality are passed through the variational IB encoder we designed to obtain the hidden layer features $H_1^i, H_2^i, \ldots, H_{L-1}^i$ and the compressed representation $H_L^i$, where $L$ is the number of encoder layers and $H_L^i$ represents the features output by the encoder of the $i$th modality at the $L$th layer. Although encoders of different modalities have the same structure, their training processes are independent of each other. 2) Contrastive learning loss: By comparing features $H_1^i, H_2^i, \ldots, H_{L-1}^i$ between different modalities and the clustering results $Y^i$, the commonalities between different modalities are explored to learn features $H_L^i$ that are more conducive to clustering. 3) Clustering layer: The clustering result $Y^i$ of $i$-th modality is obtained directly through the compressed feature $H_L^i$ through a linear layer and a softmax function.

### 2.4. Hidden-layer Information

After the original feature $X^i$ enters the encoder, it will be gradually compressed into a more compact representation $H_L^i$. The hidden layer representation $H_1^i, H_2^i, ..., H_{L-1}^i$ is the intermediate result of the compression behavior. Explicitly introducing reasonable hidden layer representations $H_1^i$, $H_2^i, ..., H_{L-1}^i$ into the objective function has the following advantages:

- **Richer feature representations:** As the network depth increases, each hidden layer gradually refines the feature representation. This makes the representation more informative and structured, which is particularly beneficial for clustering tasks.

- **Enhanced model interpretability:** Hidden layer representations provide insight into how the model gradually transforms the input. This transparency helps understand the learned hierarchical structure, facilitating model analysis and fine-tuning.

- **Optimization benefits of the information bottleneck framework:** Introducing hidden layer information in the information bottleneck framework helps regulate the trade-off between compression and retention of relevant information, resulting in more discriminative and generalizable clustering results.

### 2.5. Objective Function

**Information Bottleneck**. IB compresses the original features $X^i$ to obtain a good compression representation $H_L^i$, so as to obtain a better clustering result $Y^i$. The hidden layer information $H_1^i, H_2^i, ..., H_{L-1}^i$ can participate in this

compression process, helping us to obtain a better compression representation $H_L^i$. Therefore, Our method explicitly introduces the hidden layer information into IB and obtains the following loss function:

$$\mathcal{L}_1 = \sum_{i=1}^{v} \sum_{l=1}^{L} I(X^i; H_l^i) - \beta I(Y^i; H_l^i) \qquad (3)$$

where $I(X^i; H_l^i)$ is the mutual information between the original feature $X^i$ and the feature $H_l^i$, $I(Y^i; H_l^i)$ is the mutual information between the feature $H_l^i$ and the clustering result $Y^i$, and the parameter $\beta$ is used to control the compression and retention of feature information. The smaller $\beta$, the higher the degree of information compression.

**Consistency Information.** In MMC tasks, it is a common method to explore the consistency information between multiple modalities to seek reasonable data parition. In order to maximize the use of the consistency information between different modalities, we not only consider the consistency information at the feature level, but also the consistency information at the cluster level. At the same time, in order to ensure that the compressed features $H_L^i$ of different modalities are as consistent as possible, the proposed SDCIB keeps the feature information $H_1^i$, $H_2^i$, ..., $H_{L-1}^i$ of different modalities consistent as much as possible in the hidden layer. Therefore, the loss function of the proposed SDCIB in terms of consistency information is:

$$\mathcal{L}_2 = \sum_{i=1}^{v} \sum_{j=1}^{v} \mathbb{I}_{i \neq j}[I(Y^i; Y^j) + \sum_{l=1}^{L} I(H_l^i; H_l^j)] \quad (4)$$

where $I(H_l^i; H_l^j)$ is the mutual information between features $H_l^i$ and $H_l^j$ of different modalities, which measures the consistency of $H_l^i$ and $H_l^j$. $I(Y^i; Y^j)$ is the mutual information between the clustering results $Y^i$ and $Y^j$ of different modalities, representing the consistency of $Y^i$ and $Y^j$.

**Overall Objective Function.** Therefore, the proposed SDCIB can compress the original features while maintaining the consistency between modalities and the correlation between the compressed features and data partitions. Moreover, compared with the ordinary deep IB method, it further explicitly utilizes the hidden layer in the encoder. Therefore, we call it Super Deep Contrastvie Inforamtion Bottleneck. Its overall loss function is as follows:

$$\begin{aligned} \mathcal{L}_{SDCIB} &= \alpha \mathcal{L}_{SIB} - (1-\alpha)\mathcal{L}_{Con} \\ &= \alpha \sum_{i=1}^{v} \sum_{l=1}^{L} [I(X^i; H_l^i) - \beta I(Y^i; H_l^i)] \\ &- (1-\alpha) \sum_{i=1}^{v} \sum_{j=1}^{v} \mathbb{I}_{i \neq j}[I(Y^i; Y^j) + \sum_{l=1}^{L} I(H_l^i; H_l^j)] \end{aligned}$$
(5)

where $\alpha$ is used to weigh the importance of IB and consistency information in obtaining compressed features.

## 2.6. Optimization

In high-dimensional continuous space, the calculation of mutual information has always been a difficult problem. In order to obtain a good representation of mutual information, we adopted three different strategies to estimate mutual information, namely the variational method, Information Noise-Contrastive Estimation (InfoNCE)(Song & Ermon, 2020) and Mutual Information Neural Estimation (MINE) (Belghazi et al., 2018).

**Variational method** is used to calculate the amount of information $I(X^i; H_l^i)$ retained by the compressed variable $H_l^i$ of the original feature $X^i$. From the definition of mutual information, we can get the calculation formula of mutual information as follows:

$$\begin{aligned} I(X^i; H_l^i) &= \int_{x^i, h_l^i} p(x^i, h_l^i) log \frac{p(x^i, h_l^i)}{p(x^i)p(h_l^i)} \\ &= \int_{x^i, h_l^i} p(x^i, h_l^i) log \frac{p(h_l^i|x^i)}{p(h_l^i)} \end{aligned}$$
(6)

Since $p(h_l^i)$ is difficult to obtain, we use $q(h_l^i)$ to approximate $p(h_l^i)$. According to the non-negativity of Kullback-Leibler (KL) divergence, we can get:

$$\begin{aligned} KL[p(h_l^i)|q(h_l^i)] &= \int_{h_l^i} p(h_l^i) log \frac{p(h_l^i)}{q(h_l^i)} > 0 \\ &\Rightarrow \int_{h_l^i} p(h_l^i) log\, p(h_l^i) > \int_{h_l^i} p(h_l^i) log\, q(h_l^i) \end{aligned}$$
(7)

Rewrite $I(X^i; H_l^i)$ into the following inequality form:

$$\begin{aligned} I(X^i; H_l^i) &= \int_{x^i, h_l^i} p(x^i, h_l^i) log \frac{p(h_l^i|x^i)}{p(h_l^i)} \\ &< \int_{x^i, h_l^i} p(x^i, h_l^i) log \frac{p(h_l^i|x^i)}{q(h_l^i)} \\ &< \int_{x^i, h_l^i} p(h_l^i|x^i)p(x^i) log \frac{p(h_l^i|x^i)}{q(h_l^i)} \end{aligned}$$
(8)

To simplify the formula, we use Monte Carlo sampling (Von Ahn & Dabbish, 2005) to replace $p(x^i)$ with $\frac{1}{N}$ and get:

$$I(X^i; H_l^i) < \frac{1}{N} \sum_{i}^{N} \int_{h_l^i} p(h_l^i|x^i)p(x^i) log \frac{p(h_l^i|x^i)}{q(h_l^i)} \quad (9)$$

Where N is the number of data samples.

Assuming that $p(h_l^i|x^i)$ obeys a Gaussian distribution, its mean $\mu$ and standard deviation $\sigma$ can be obtained by the variational IB encoder, where $\epsilon$ obeys a standard normal

**Algorithm 1** Algorithm for Optimizing the proposed SD-CIB
___
1: **Input:** Dataset with $M$ modalities $\{X^i\}_{i=1}^m$, the number of clusters $K$, the parameter $\alpha$, $\beta$.
2: **Output:** Final partition $Y$.
3: **Random Initialization:** Randomly initialize the parameters of $M$ modality-specific encoders and $M * L$ mutual information estimators.
4: **repeat**
5:      Calculate $L_{MINE}$ by Eq. (12)
6:      Optimize the parameters of mutual information estimators
7:      Obtain $I(Y^i; H_l^i)$ by mutual information estimators
8:      Calculate $I(X^i; H_l^i)$ by Eq. (8)
9:      Calculate $I(H_l^i; H_l^j)$ and $I(Y^i; Y^j)$ and by Eq. (11)
10:      Optimize the parameters of modality-specific encoders
11: **until** Convergence
12: Obtain partition $Y^i$
13: Obtain final partition $Y$ by Eq. (2)
___

distribution. Now $I(X^i; H_l^i)$ can be rewritten as:

$$I(X^i; H_l^i) < \frac{1}{N} \sum_i^N \mathbb{E}_\epsilon log \frac{p(h_l^i|x^i)}{q(h_l^i)} \\ < \frac{1}{N} \sum_i^N \mathbb{E}_\epsilon KL[p(h_l^i|x^i), q(h_l^i)] \quad (10)$$

**InfoNCE** is used when calculating mutual information $I(H_l^i; H_l^j)$ between compressed variables of different modalities and mutual information $I(Y^i; Y^j)$ between clustering results. InfoNCE estimates the mutual information by contrastive loss:

$$I(Y^i; Y^j) \approx \mathbb{E}_{p(y^i,y^j)} \left[ \log \frac{\exp(f_\theta(y^i, y^j))}{\sum_{n=1}^N \exp(f_\theta(y^i, y_n^j))} \right] + logN \quad (11)$$

where $f_\theta$ represents the similarity, $(y^i, y^j)$ is a positive sample pair, $(y^i, y_n^j)$ is a negative sample pair, and $N$ is the total number of samples. Similarly, the mutual information $I(H_l^i; H_l^j)$ can also be calculated by Eq. (11).

**MINE** is used to calculate the amount of information that the compressed variable $H_l^i$ retains about the label $Y^i$. The dimensions of $H_l^i$ and $Y^i$ are not consistent, where the dimension of $H_l^i$ is higher than that of $Y^i$. The variational method requires manual alignment of the dimensions of $H_l^i$ and $Y^i$, which may cause information loss during the alignment process. The InfoNCE method requires the construction of a large number of negative samples, and due to the small dimension of $Y^i$, which may lead to insufficient discrimination between positive and negative samples,

thus affecting the accuracy of the estimation. In contrast, MINE provides a more robust mutual information estimation method that obtains the lower bound of the mutual information by optimizing the loss:

$$L_{\text{MINE}} = -\mathbb{E}_{p(x,y)}[T_\theta(x,y)] + \log \mathbb{E}_{p(x)p(y)}[\exp(T_\theta(x,y))] \quad (12)$$

where $T$ is the function that the neural network needs to fit, and $\theta$ is the learnable parameter.

For the specific optimization process, see Algorithm 1.

## 3. Experiment

### 3.1. Datasets

We selected 4 datasets of different scales for experimental evaluation, with detailed datasets information presented in Table 1.

**Caltech-2V**(Fei-Fei et al., 2004) contains 1,440 image samples, categorized into 7 classes based on WM and CENTRIST modalities. **Event** (Li & Fei-Fei, 2007) encompasses 1,579 sports event image samples, divided into 8 categories based on 3 modalities: Color Attention, SIFT, and TPLBP. **IAPR** (Grubinger et al., 2006) includes 7,855 image samples, accompanied by natural language descriptions, and is divided into 6 categories using SIFT representation and BoW model modalities. **ESP** (Von Ahn & Dabbish, 2005) sourced from a social image collection on an image annotation game website, comprises 11,032 image samples, categorized into 7 classes.

### 3.2. Compared Methods

To evaluate the effectiveness of the proposed SDCIB, we adopt fourteen baseline methods as comparisons. Fourteen of these methods span three major categories of modality clustering approaches.

**Single-Modal Clustering / Full-Modal Clustering Methods**: K-Means (KM) and Normalized Cuts (Ncuts) are two classic and widely used single-modal clustering methods. Based on single-modal clustering, full-modal versions are developed by splicing multiple modalities into a unified representation, ultimately leading to All-modal K-Means (AmKM) and All-modal Ncuts (AmNcuts).

**Traditional MMC Methods**: Below are 4 representative traditional MMC methods. These methods enhance the

*Table 1.* Details of various kinds of multi-modal datasets.

| DATASET | MODALITIES | SAMPLES | CLUSTERS | DIMENSIONALITY |
|---|---|---|---|---|
| CALTECH-2V | 2 | 1440 | 7 | (40,254) |
| EVENT | 3 | 1579 | 8 | (1000,1000,1000) |
| IAPR | 2 | 7855 | 6 | (1200,500) |
| ESP | 3 | 11032 | 7 | (300,300,300) |

robustness of clustering from different methodological perspectives.

(1) CoregMVSC (Kumar et al., 2011):A multi-modal spectral clustering method that applies co-regularization to the clustering results.

(2) RMKMC (Cai et al., 2013): A multi-modal k-means clustering method that adaptively adjusts modality weights to handle differences in modal quality.

(3) SwMC (Nie et al., 2017): A totally self-weighted multi-modal clustering method for automatic modality weighting.

(4) ONMSC (Zhou et al., 2020): A multi-modal clustering method that integrates the neighborhood information of first-order and high-order Laplacian matrices.

**Deep MMC Methods**: Below are 6 representative state-of-the-art methods in the field of deep MMC, demonstrating significant progress in recent years.

(1) SiMVC and CoMVC (Trosten et al., 2021): The paper introduces two multi-modal clustering models. SiMVC is a simple baseline model for deep clustering that performs clustering by learning a weighted fusion of representations from different modalities via linear combination. CoMVC builds on this by introducing a contrastive alignment module to overcome the limitations of traditional alignment methods.

(2) MFLVC (Xu et al., 2022): A hierarchical feature learning clustering method that efficiently integrates multi-level feature learning and contrastive learning.

(3) DealMVC (Yang et al., 2023): A clustering method that ensures the consistency of similar samples using a dual contrastive calibration network.

(4) ICMVC (Chao et al., 2024): An end-to-end clustering method that handles missing data through multi-modal consistency transfer and graph convolutional networks, and combines contrastive learning.

(5) DIVICE (Lu et al., 2024): A multi-modal clustering method based on decoupled contrastive learning and high-order random walks, and integrates the idea of contrastive learning to improve clustering performance.

### 3.3. Settings of Experiments

The experiments adopts two commonly used clustering evaluation metrics, accuracy (ACC) and normalized mutual information (NMI), to assess clustering performance. Higher values of these metrics indicate better clustering quality. For single-modal clustering methods, the clustering result of the best-performing modality is selected as the final result. For other MMC methods, the parameter configurations in our experiments follow the original settings provided in their respective papers, and the optimal results under the recommended parameters are chosen for comparison.

In the implementation of the proposed SDCIB, parameters $\alpha$ and $\beta$ are defined. $\alpha$ takes values from the range $\{0.1, 0.2, 0.4, 0.6, 0.8\}$, and $\beta$ takes values from the range $\{1, 10, 100, 1000, 10000\}$. The specific choices of $\alpha$ and $\beta$ will be presented in detail in subsequent chapters. The entire training process of the experiment is completed within 40 epochs, with a batch size of 32. The proposed SDCIB consists of $M$ modality-specific encoders, $4*M$ mutual information estimators, and $M$ clustering layers. Each

*Table 2.* Clustering performance with Acc and NMI on various kinds of datasets (the bold and underlined values in the table represent the best and second-best results respectively).

| METHODS | CALTECH-2V | | EVENT | | IAPR | | ESP | |
|---|---|---|---|---|---|---|---|---|
| | ACC | NMI | ACC | NMI | ACC | NMI | ACC | NMI |
| KM | 41.6 | 30.5 | 34.7 | 20.7 | 38.9 | 17.2 | 48.4 | 33.5 |
| NCUTS (TPAMI'00) | 39.9 | 31.2 | 34.8 | 15.5 | 41.9 | 18.9 | 45.7 | 29.8 |
| AMKM | 46.4 | 31.4 | 28.7 | 11.6 | 40.4 | 17.0 | 35.0 | 20.7 |
| AMNCUTS (TPAMI'00) | 42.8 | 25.2 | 35.2 | 20.3 | 42.2 | 18.9 | 32.5 | 19.0 |
| COREGMVSC (NIPS'11) | 49.2 | 39.6 | 35.5 | 22.2 | 35.1 | 18.4 | 45.2 | 30.7 |
| RMKMC (IJCAI'13) | 51.4 | 33.5 | 39.5 | 25.1 | 36.4 | 15.9 | 35.1 | 20.8 |
| SWMC (IJCAI'17) | 34.2 | 26.6 | 16.7 | 2.2 | 30.2 | 23.1 | 37.8 | 22.8 |
| ONMSC (AAAI'20) | 34.2 | 26.6 | 48.6 | 33.8 | 21.6 | 11.1 | 21.2 | 12.2 |
| SIMVC (CVPR'21) | 51.1 | 36.9 | 36.8 | 23.1 | 42.7 | 18.5 | 33.6 | 14.6 |
| COMVC (CVPR'21) | 59.2 | 49.2 | 49.1 | 35.5 | 46.7 | 21.5 | 43.4 | 27.3 |
| MFLVC (CVPR'22) | 61.5 | 53.6 | 48.5 | 34.9 | 47.3 | 22.6 | 52.1 | 36.9 |
| DEALMVC (ACM MM'23) | 47.6 | 37.9 | 26.5 | 9.3 | 35.0 | 10.8 | 43.4 | 27.4 |
| ICMVC (AAAI'24) | 49.6 | 37.9 | 36.4 | 30.3 | 37.1 | 16.8 | 46.7 | 30.0 |
| DIVICE (AAAI'24) | 64.1 | 52.9 | 31.4 | 12.4 | 45.6 | 23.0 | 47.2 | 28.8 |
| SDCIB | **67.5** | **59.2** | **56.5** | **36.4** | **52.9** | **28.7** | **61.4** | **44.7** |
| OURS VS BEST COMPARED | 3.4↑ | 5.6↑ | 7.4↑ | 0.9↑ | 5.6↑ | 5.6↑ | 9.3↑ | 7.8↑ |

modality-specific encoder contains 4 fully connected layers with dimensions of 1024, 1024, 1024, and 128, respectively. Each fully connected layer is followed by a BatchNorm layer for representation normalization and a ReLU layer as the activation function. MINE continuously optimizes the correlation between two features to obtain a more accurate mutual information measure, laying a solid foundation for subsequent clustering. The clustering layer consists of a fully connected layer and a softmax layer to obtain the final clustering results. Meanwhile, we use the Adam optimizer for parameter optimization, with an initial learning rate set to 0.0001.

### 3.4. Results and Analysis

In this chapter, we conduct comprehensive experiments on 4 datasets of different sizes, with the ACC and NMI values for fourteen comparison methods and the proposed SDCIB are listed in Table 2. In the specific numerical comparison, the proposed SDCIB demonstrates strong competitiveness.

- Compared to single-modal clustering and full-modal clustering methods, the proposed SDCIB significantly outperforms in clustering performance. Specifically, on the Caltech-2V dataset, the ACC metric of the proposed SDCIB exceeds the single-modal KM method and Ncuts method by 25.9% and 27.6%, respectively. The ESP dataset also outperforms the full-modality AmKM and AmNcuts methods in terms of the NMI metric by 24.0% and 25.7%, in turn. Meanwhile, compared to the performance of single-modal and full-modal methods, there is no significant advantage in numerical terms. This is because in single-modal clustering, the optimal modal clustering result is selected,

which does not fully represent the entire dataset.

- Compared to traditional MMC methods, the proposed SDCIB demonstrates a clear advantage in clustering performance. For example, on the Caltech-2V and IAPR datasets, the proposed method improves the best ACC results of traditional methods by 16.1% and 16.5%, respectively. At the same time, traditional MMC methods generally outperform full-modal methods, which suggests that the simple full-modal approach, formed by splicing individual modalities, does not properly apply the information within the modality.

- Compared to deep MMC methods, the proposed SDCIB still demonstrates outstanding clustering performance. For example, on the Event dataset, the proposed method outperforms the deep MMC methods SiMVC, CoMVC, MFLVC, DealMVC, ICMVC, and DIVICE in ACC by values of 19.7%, 7.4%, 8.0%, 30.0%, 20.1%, and 25.1%. This is attributed to our approach of fully leveraging the internal information within the data and innovatively using hidden layer information to assist in clustering, thereby better exploiting the advantages of deep learning. Additionally, the proposed SDCIB incorporates the information bottleneck principle and introduces contrastive learning to jointly guide the clustering process.

### 3.5. Parameter Selection

**Hyperparameters $\alpha$ and $\beta$.** We analyze the the parameters $\alpha$ and $\beta$ of the proposed SDCIB. The values of parameter $\alpha$ are $\{0.1, 0.2, 0.4, 0.6, 0.8\}$, and the values of $\beta$ are $\{1, 10, 100, 1000, 10000\}$. The parameters form all possible combinations through the Cartesian product. For each

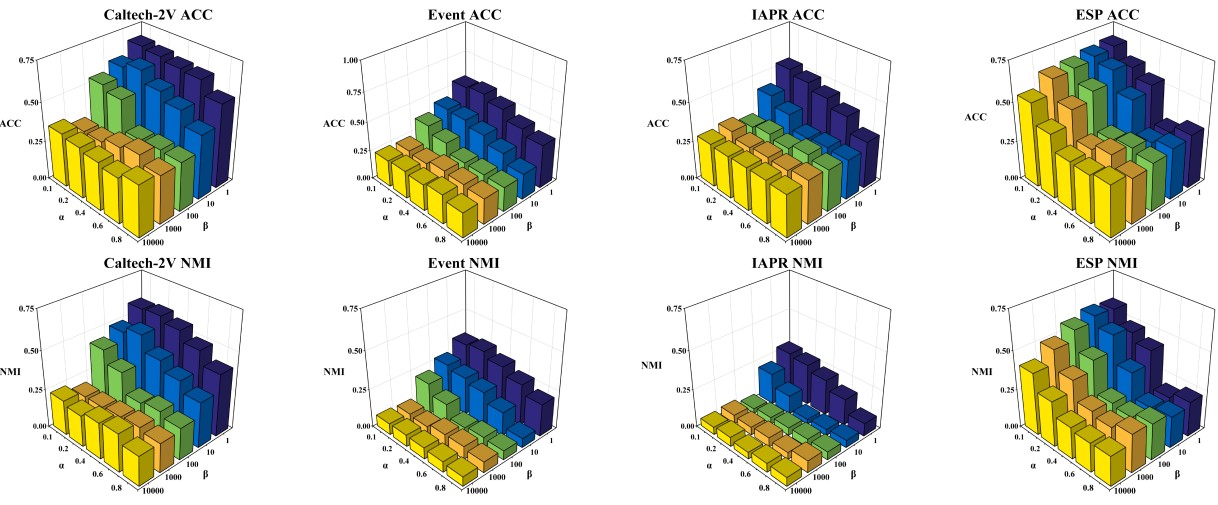

*Figure 2.* The parameter analysis of our extended the proposed SDCIB on the Caltech-2V, Event, IAPR and ESP datasets is presented, with each bar in the figure corresponding to a specific pair of $(\alpha, \beta)$ values.

*Table 3.* Model network layer evaluation on Caltech-2V and Event datasets (the bold values in the table represent the best).

| HIDDEN LAYERS | CALTECH-2V | | EVENT | | IAPR | | ESP | |
|---|---|---|---|---|---|---|---|---|
| | ACC | NMI | ACC | NMI | ACC | NMI | ACC | NMI |
| 2 | 62.6 | 54.8 | 50.1 | 30.3 | 52.7 | 27.2 | **62.1** | 42.5 |
| 3 | 64.4 | 54.2 | 55.0 | **36.6** | 51.8 | 28.0 | 61.6 | **44.7** |
| 4 | **67.5** | **59.2** | **56.5** | 36.4 | **52.9** | 28.7 | 61.4 | **44.7** |
| 5 | 65.7 | 58.3 | 47.1 | 30.3 | 49.4 | 26.1 | 58.8 | 42.1 |
| 6 | 64.5 | 57.1 | 50.5 | 31.3 | 52.4 | **38.5** | 62.0 | 43.8 |
| 7 | 65.3 | 58.3 | 44.5 | 27.6 | 47.6 | 27.3 | 61.2 | 43.7 |

*Table 4.* Ablation results of the Caltech-2V, Event, IAPR, and ESP datasets (the bold values in the table represent the best).

| METHODS | CALTECH-2V | | EVENT | | IAPR | | ESP | |
|---|---|---|---|---|---|---|---|---|
| | ACC | NMI | ACC | NMI | ACC | NMI | ACC | NMI |
| REMOVING $I(Y^i; Y^j)$ | 41.8 | 33.2 | 30.3 | 15.5 | 38.5 | 20.1 | 36.3 | 27.6 |
| REMOVING $I(H_l^i; H_l^j)$ | 63.1 | 52.2 | 45.0 | 25.3 | 48.8 | 23.5 | **62.3** | 43.4 |
| REMOVING $I(Y^i; Y^j)$ AND $I(H_l^i; H_l^j)$ | 32.2 | 23.6 | 21.6 | 9.1 | 30.0 | 6.8 | 27.1 | 16.3 |
| SDCIB | **67.5** | **59.2** | **56.5** | **36.4** | **52.9** | **28.7** | 61.4 | **44.7** |

set of parameters, we record the ACC and NMI values, as shown in Figure 2. It is observed that when the parameters $\alpha = 0.1$ and $\beta = 1$, the ACC and NMI values across various datasets generally outperform other parameter settings. This indicates that the proposed SDCIB demonstrates universality and stability under these parameters. The combination of $\alpha = 0.1$ and $\beta = 1$ effectively balances the key factors within the method, highlighting the effectiveness and rationality of the proposed SDCIB design.

**Number of encoder layers.** The hidden layers in the proposed SDCIB represent a highly ingenious design. To investigate the impact of the number of network layers on clustering performance, we conducted experiments on the Caltech-2V, Event, IAPR, and ESP datasets, varying the number of network layers to 2, 3, 4, 5, 6, and 7. The experimental results are shown in Table 3. The results indicate that having more or fewer layers does not necessarily lead to better performance. Notably, the optimal results for the four datasets are generally achieved with a 4-layer network. Therefore, in the architecture of the proposed SDCIB, we adopt a 4-layer network with dimensions set to 1024, 1024, 1024, and 128.

### 3.6. Ablation Study

In this chapter, we conduct ablation studies to evaluate the contribution of different components in the objective function of the proposed SDCIB. Specifically, two terms are subject to removal: (1) mutual information between $Y^i$ and $Y^j$, denoted as $I(Y^i; Y^j)$, and (2) mutual information between $H_l^i$ and $H_l^j$, denoted as $I(H_l^i; H_l^j)$. Table 4 presents the results of the ablation experiments, leading to the following conclusions.

First, when we remove term (1), the ACC and NMI values for the 4 datasets decrease significantly, especially for the Event dataset, where the ACC value drops by 26.2%. This is mainly because the $I(Y^i; Y^j)$ term considers clustering information between different modalities, which is beneficial for modality fusion. Second, when we remove term (2), both ACC and NMI show a noticeable decline, except for the ESP dataset, where the ACC value increases slightly by 0.9% compared to the proposed SDCIB. Finally, when (1) and (2) are deleted at the same time, the clustering performance of the four datasets is almost halved. For example, on the Caltech-2V dataset, the ACC drops by 35.3%. The above experiments validate the significant contribution of each component in the proposed SDCIB to the final clustering performance, fully proving its effectiveness.

### 3.7. Necessity of hidden layer information

To verify the necessity of hidden layer information, we conducted experimental investigations in this chapter. Under the condition that the parameter settings are consistent, we designed two groups of experiments. One experiment ignores the information from the hidden layers and focuses only on the final output layer $H_L^i$, while the other experiment retains the hidden layer information extraction part of the proposed SDCIB. The specific experimental results are presented in Table 5, where it can be observed that the proposed SDCIB demonstrates varying degrees of improvement across different datasets. Notably, on the Event dataset, the ACC value increased by 10.6%, and the NMI value improved by 7.3%. The experiments provide strong evidence

*Table 5.* Experimental results on the Caltech-2V, Event, IAPR, and ESP datasets with and without the application of hidden layer information (the bold values in the table represent the best).

| METHODS | CALTECH-2V | | EVENT | | IAPR | | ESP | |
|---|---|---|---|---|---|---|---|---|
| | ACC | NMI | ACC | NMI | ACC | NMI | ACC | NMI |
| SDCIB-NO-HIDDEN | 63.6 | 55.8 | 45.9 | 29.1 | 48.9 | 26.9 | 57.9 | 40.0 |
| SDCIB | **67.5** | **59.2** | **56.5** | **36.4** | **52.9** | **28.7** | **61.4** | **44.7** |
| IMPROVEMENT | 3.9↑ | 3.4↑ | 10.6↑ | 7.3↑ | 4.0↑ | 1.8↑ | 3.5↑ | 4.7↑ |

for the crucial role of hidden layer information. Moreover, they indicate that fully mining and utilizing hidden layer information helps to more deeply explore the intrinsic relationships and latent structures among modalities, thereby enhancing the accuracy and robustness of clustering results.

### 3.8. Convergence Analysis

To study the convergence of the proposed SDCIB, we present the total loss function variation curves of the Caltech-2V and IAPR datasets in a chart. As shown in Figure 3, We observe its loss value, which shows an overall downward trend. At about 500 epochs, the function basically converges to a relatively stable value. Therefore, the proposed SDCIB demonstrates rapid convergence within a certain range, which not only validates the effectiveness of the proposed SDCIB but also demonstrates the reliability and stability of the proposed SDCIB.

### 3.9. Visualization Validation

In this chapter, we use t-distributed stochastic neighbor embedding (t-SNE) to visualize the Caltech-2V, Event, IAPR, and ESP datasets. As shown in Figure 4 different colors represent different clusters. It is evident that the proposed SDCIB shows a more compact distribution of data samples within the same cluster, while the samples from different clusters are more dispersed. Overall, the distribution is clearer and more intuitive. This visualization directly demonstrates the advantages of the proposed SDCIB clustering performance, highlighting its ability to capture fine-grained patterns in complex datasets.

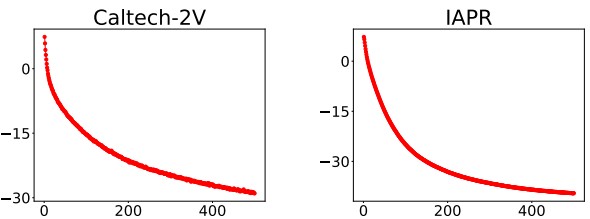

*Figure 3.* The loss function variation curves of the Caltech-2V and IAPR datasets.

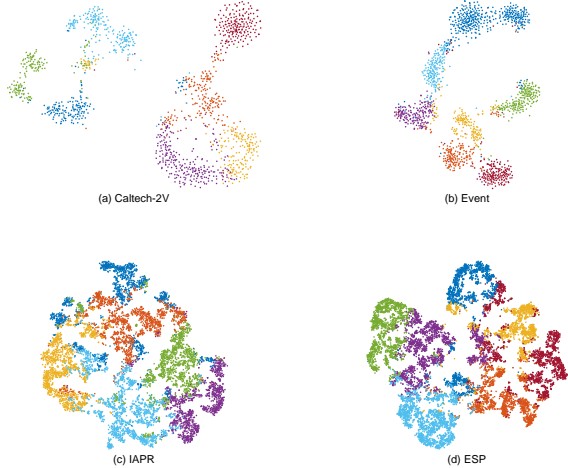

*Figure 4.* Visualization of the Caltech-2V, Event, IAPR, and ESP datasets (different colors represent different clusters).

## 4. Conclusion

This paper innovatively proposes the super deep contrastive information bottleneck (SDCIB) for MMC method to address the challenges currently faced by MMC, enabling a more thorough extraction of data information. Specifically, the proposed SDCIB not only incorporates the rich information contained in the encoder's hidden layers into the clustering process, but also performs dual optimization from two consistency information perspectives: feature distribution and clustering assignment. Compared to fourteen advanced MMC methods, the superiority of the proposed SDCIB has been repeatedly validated on the Caltech-2V, Event, IAPR, and ESP datasets.

However, the proposed SDCIB still faces several challenges, such as limited performance on incomplete data samples (Liu et al., 2024), reliance on a predetermined number of clusters, and the use of a batch learning strategy. These factors may limit the flexibility of the proposed SDCIB in practical applications, especially in complex scenarios involving unknown structures, missing data, or real-time requirements. Addressing these issues will be the focus of our future work.

## Acknowledgements

The authors thank anonymous reviewers for their constructive comments. This work was supported by National Natural Science Foundation of China under Grant 62206254, Henan Province Outstanding Youth Science Fund Program under Grant 252300421223 and China Postdoctoral Science Foundation under Grant 2024T170843 and 2023M743186.

## Impact Statement

This paper presents work whose goal is to advance the field of Machine Learning. There are many potential societal consequences of our work, none which we feel must be specifically highlighted here.

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
