# OpenReview forum: "Super Deep Contrastive Information Bottleneck for Multi-modal Clustering"
_ICML.cc/2025/Conference — ICML 2025 poster_

### Official Review · Reviewer_kMYo · 2025-03-08

**Overall Recommendation:** 3

**Summary:**

To fully explore the complex latent information and interdependencies among multi-modal data, this paper propose a super deep contrastive information bottleneck for multi-modal clustering method. It incorporates the rich information from the hidden layers of the encoder into the clustering process to comprehensively capture modality features and their associations. Furthermore, a dual contrastive learning strategy is designed to ensure more precise and stable clustering performance. Experimental results show the promising performance by the method, which fully validate its advantages and well-designed framework.

**Claims And Evidence:**

The claims made in the submission are supported by clear and convincing evidence in the experimental parts in section 3.

**Essential References Not Discussed:**

There is no essential references not discussed in this paper.

**Experimental Designs Or Analyses:**

I have checked the soundness/validity of any experimental designs or analyses in the experimental sections.

**Methods And Evaluation Criteria:**

The proposed methods and evaluation criteria make sense for the problem in this paper. This paper propose a super deep contrastive information bottleneck for multi-modal clustering method, and the experimental results show the promising performance by the method, which fully validate its advantages and well-designed framework.

**Other Comments Or Suggestions:**

No other comments or suggestions.

**Other Strengths And Weaknesses:**

This paper propose a novel super deep contrastive information bottleneck for multi-modal clustering, which is new and well-organized. The experiments are well-designed and have shown its advantages over existing methods. However, there are still some concerns shown below:
1.Hidden layer information integration mechanism: Is it a shared encoder or an independent encoder? How to fuse the hidden layer information of different modalities?
2.Improper language expression: The lengthy sentence structure affects readability.
3.The experimental data is not public: it is recommended to provide a data download link to improve the reproducibility of the paper.
4. Shortcoming is missing: The shortcomings of the proposed method are not given in the paper. Generally, for a conference paper, advantage and limitation analysis are the major parts.

**Questions For Authors:**

Please see the above.

**Relation To Broader Scientific Literature:**

This paper proposes a novel super deep contrastive information bottleneck for multi-modal clustering method, and it is different from prior related findings.

**Theoretical Claims:**

There is no theoretical claims in this paper.

---

> ### Author Rebuttal · Authors · 2025-03-31
>
> Thank you for the insightful comments and constructive suggestions. We have carefully revised the whole manuscript and provided detailed responses to each point below.
>
> **Q1: Hidden layer information integration mechanism: Is it a shared encoder or an independent encoder? How to fuse the hidden layer information of different modalities?**
> ***Response：*** Thank you for the insightful question. Below, we clarify the integration mechanism of hidden layer information in our approach.
> Instead of a shared encoder, we adopt an independent encoder for each modality. Different modalities usually have different statistical properties, making it difficult to directly fuse their hidden layer information at an early stage. This design allows each modality to learn modality-specific feature representations, which is beneficial given the different statistical properties of various modalities.
> Instead of performing direct fusion at the hidden layer level, we compare the hidden layer representations of different modalities to extract shared and complementary information. Specifically, each encoder learns to align its representations by evaluating the relationship between its hidden layer features and the hidden layer features of other modalities. We achieve this by maximizing the mutual information of hidden layer features between different modalities, which encourages aligned representations while preserving modality-specific features.
> By adopting this strategy, we ensure that the final multimodal representations are both consistent and rich, leading to a more efficient feature learning process. We hope this explanation clarifies our integration mechanism.
>
> **Q2: Improper language expression: The lengthy sentence structure affects readability.**
> ***Response：*** Thank you for the constructive comments on sentence structure. To enhance the readability of the manuscript, we have made extensive revisions. We focused on checking redundant sentences and reducing them through segmentation or conciseness to improve clarity. For example, the sentence in the *'Introduction'* section, *'The proposed SDCIB can efficiently and meticulously mine the latent information between modalities through the hidden layers, and simultaneously focus on both feature distribution and clustering assignment to better capture the inherent structure of the data,'* has been revised to: *'The proposed SDCIB efficiently mines the latent information between modalities through the hidden layers. Meanwhile, it simultaneously considers both feature distribution and clustering assignment to better capture the inherent structure of the data.'* In the final version, we will adjust lengthy expressions into clearer and more readable sentences.
>
> **Q3: The experimental data is not public: it is recommended to provide a data download link to improve the reproducibility of the paper.**
> ***Response：*** We appreciate your suggestion and realize the importance of reproducibility in scientific research. Currently, our experimental section already cites the original source of the dataset to provide transparency of its origin. However, we also understand that providing a direct download link can further improve the accessibility of readers. We will revise the manuscript to add specific download links to ensure that it is publicly available and reproducible.
>
> **Q4: Shortcoming is missing: The shortcomings of the proposed method are not given in the paper. Generally, for a conference paper, advantage and limitation analysis are the major parts.**
> ***Response：*** Thank you for the insightful comment. Based on your suggestion, we have updated the conclusion to include the following limitations of the proposed SDCIB:
> * The proposed SDCIB demonstrates limited effectiveness when dealing with incomplete data, particularly when certain modalities or features are missing. In these cases, the model may face difficulties in accurately capturing the relationships between modalities, which could result in less optimal clustering outcomes.
> * The method requires the number of clusters to be predetermined, which may be a limitation as it assumes prior knowledge of the data's structure, making it less flexible in scenarios where this information is unavailable.
> * The proposed SDCIB primarily relies on batch learning, which may not be suitable for certain applications, such as streamed multi-modal data.
>
> While we recognize that these limitations could impact certain scenarios, we believe that future research can explore ways to address them, with the goal of improving the method’s robustness and expanding its range of applications.
>
> Thanks again for the valuable suggestions provided by the reviewer. The modifications will be added to the final version.

---

### Official Review · Reviewer_aWGf · 2025-03-09

**Overall Recommendation:** 5

**Summary:**

This paper proposes a Super Deep Contrastive Information Bottleneck (SDCIB) for multi-modal clustering, designed to fully exploit the latent information in multi-modal data. SDCIB integrates the rich information from the hidden layers of the encoder into the clustering process, optimizing both feature distribution and clustering assignments through contrastive learning. Experimental results on four multi-modal datasets demonstrate that SDCIB outperforms existing approaches.

**Claims And Evidence:**

The claims made in the submission are well-supported by clear and convincing evidence. The experimental results on multiple multi-modal datasets consistently demonstrate that the SDCIB method outperforms existing approaches.

**Essential References Not Discussed:**

Critical references have been included.

**Experimental Designs Or Analyses:**

I have reviewed the soundness and validity of the experimental designs and analyses in Section 3, including subsections 3.1 to 3.9. The experimental setup, dataset selection, evaluation metrics, and comparative analysis are appropriately designed to support the paper's claims.

**Methods And Evaluation Criteria:**

The proposed methods and evaluation criteria are well-suited for the multi-modal clustering problem. SDCIB effectively leverages the rich information from the encoder's hidden layers, optimizing both feature distribution and clustering assignments through contrastive learning, which aligns to improve clustering performance in multi-modal settings.

**Other Comments Or Suggestions:**

I have one suggestion regarding the abstract: the authors are encouraged to provide quantitative descriptions of the results. For example, "The experiment shows that the method is effective" should be changed to "The accuracy of the method on the X dataset is improved by 5%".

**Other Strengths And Weaknesses:**

The paper presents a well-structured argument and is written with clarity. Below are my detailed comments:
1. The paper discusses the use of deeper hidden layers to explore relationships, but the specific advantages of this approach are not explicitly outlined. I recommend the authors provide a more detailed explanation of the benefits of using deeper layers in Section 2.
2. The authors utilize MINE to estimate mutual information. It would be helpful if the paper included a discussion of why MINE was chosen over other potential estimation methods. Were alternative methods considered and, if so, why were they not chosen?
3. There are minor formatting and writing inconsistencies that could be addressed to improve the paper. For instance, ensure consistency in the usage of terms like "Information" on Page 3, and correct the reference to "IJCAL" in the bibliography.

**Questions For Authors:**

Please refer to the above comments.

**Relation To Broader Scientific Literature:**

This paper proposes a Super Deep Contrastive Information Bottleneck (SDCIB) method for multi-modal clustering in the broader scientific literature. It improves upon existing ideas in multi-modal clustering, contrastive learning, and information bottleneck, offering more efficient and powerful clustering solutions for multi-modal data.

**Theoretical Claims:**

The paper does not present any formal proofs or theoretical claims.

---

> ### Author Rebuttal · Authors · 2025-03-31
>
> Thank you for the insightful comments and constructive suggestions.
>
> **Q1: The paper discusses the use of deeper hidden layers to explore relationships, but the specific advantages of this approach are not explicitly outlined.**
> ***Response：*** Thank you for the insightful comment. The use of hidden-layer information provides the following advantages:
> * Richer Feature Representations: As the depth of the network increases, each hidden layer progressively refines the feature representations, capturing both low-level and high-level structures. This enables a more informative and structured representation, which is particularly beneficial for clustering tasks.
> * Improved Feature Compression: Deeper layers allow for a more effective compression process, where redundant information is filtered out while preserving essential details. By leveraging intermediate hidden-layer information, the model learns a more compact and meaningful feature representation.
> * Optimization Benefits in the Information Bottleneck Framework: Introducing hidden-layer information in the IB framework helps regulate the trade-off between compression and retention of relevant information, leading to more discriminative and generalizable clustering results.
>
> We will incorporate a more detailed discussion of these aspects in Section 2 to clarify the advantages of using deeper hidden layers.
>
>
> **Q2: The authors utilize MINE to estimate mutual information. It would be helpful if the paper included a discussion of why MINE was chosen over other potential estimation methods. Were alternative methods considered and, if so, why were they not chosen?**
> ***Response：*** Thank you for the suggestion. During the experiment, we did consider other methods such as variational mutual information estimation and InfoNCE. However, for mutual information $I(A;B)$, $A$ is a feature and $B$ is the label of the cluster, which results in the dimension of $A$ being much larger than that of $B$. We found that:
> * The variational method requires manual alignment of the dimensions of $A$ and $B$, which may cause information loss during the alignment process.
> * The InfoNCE method requires the construction of a large number of negative samples, and due to the small dimension of $B$, we found that this may lead to insufficient discrimination between positive and negative samples, thus affecting the accuracy of the estimation.
>
> In contrast, MINE provides a more robust mutual information estimation method, especially when there is a significant dimensional difference between the two variables. This method can estimate more efficiently and accurately without the need for dimensional alignment or worrying about insufficient discrimination of negative samples. Therefore, we chose MINE.
>
> **Q3: There are minor formatting and writing inconsistencies. For instance, ensure consistency in the usage of terms like "Information" on Page 3, and correct the reference to "IJCAL" in the bibliography.**
> ***Response：*** Thank you for the careful review. We have carefully reviewed the manuscript and made the necessary revisions. For example, the two instances of *'Information'* on page 3 had different formatting, and we have standardized them to title case as *'Hidden-layer Information'* and *'Consistency Information.'* Additionally, regarding the reference to *'IJCAL'* that you mentioned, we have corrected it in the references to: Nie, F., Li, J., and Li, X. Self-weighted multiview clustering with multiple graphs. In *Proceedings of the 26th International Joint Conference on Artificial Intelligence (IJCAI2017)*, pp. 2564–2570, 2017. Furthermore, we have checked all other references to enhance its overall precision.
>
> **Q4:The authors are encouraged to provide quantitative descriptions of the results. For example, "The experiment shows that the method is effective" should be changed to "The accuracy of the method on the X dataset is improved by 5%"**
> ***Response：*** Thank you for the detailed feedback on the content of the abstract. We have revised the abstract accordingly. Specifically, we changed the sentence *'We conduct experiments on 4 multi-modal datasets.'* to *'We conducted experiments on 4 multi-modal datasets and the accuracy of the method on the ESP dataset improved by 9.3%.  The results demonstrate the superiority and clever design of the proposed SDCIB.'* This adjustment enhances the precision of the expression and more intuitively highlights the effectiveness of the proposed SDCIB.
>
> Thanks again. The modifications will be added to the final version.

---

### Official Review · Reviewer_BTqg · 2025-03-09

**Overall Recommendation:** 3

**Summary:**

This paper proposed an information bottleneck based method named SDCIB for addressing the multi-modal clustering problem, which aims to mine the complex correlations and interdependencies among modalities. It mainly contains two aspects, first, it incorporates the different hidden layers into loss functions to fully mine the relationships among modalities. Then, it also explores the consistency information among clustering assignments of modalities. The experimental results show the superiority of the proposed method.

**Claims And Evidence:**

Yes

**Essential References Not Discussed:**

The related works are essential to understand the key contributions and no important ones are not discussed in the paper.

**Experimental Designs Or Analyses:**

I have carefully checked the soundness and validity of the experimental designs and their analysis, including all the subsections in Sec. 3.

**Methods And Evaluation Criteria:**

Yes

**Other Comments Or Suggestions:**

I have no other comments or suggestions. Please see my comments above.

**Other Strengths And Weaknesses:**

Strengths:
1. The paper presents a well-articulated motivation, with its effectiveness rigorously validated through multiple experiments.
2. This paper exhibits a notable advancement in multi-modal clustering field, demonstrating superior performance over existing methods.
3. The authors offer a comprehensive and well-structured explanation of the method, effectively highlighting its significance.


Weaknesses:
1. The full name of some abbreviations is missing, such as KL, which may influence the readability and the understanding on the paper.
2. It is seen that the improvement of the proposed method over other methods is significant. Will the source code be released to the public to enhance the development of the multi-modal clustering community？
3. Are there any limitations of the proposed method? The authors are encouraged to give them in the conclusion.

**Questions For Authors:**

I have no other questions for authors, all my questions are shown Weaknesses.

**Relation To Broader Scientific Literature:**

This paper proposed an information bottleneck based method named SDCIB for addressing the multi-modal clustering problem. I find the above is the key contributions, and is not proposed in prior findings or results.

**Theoretical Claims:**

No proofs and theoretical claims in this paper.

---

> ### Author Rebuttal · Authors · 2025-03-31
>
> Thank you for the insightful comments and constructive suggestions. We have carefully revised the whole manuscript and provided detailed responses to each point below.
>
> **Q1: The full name of some abbreviations is missing, such as KL, which may influence the readability and the understanding on the paper.**
> ***Response：*** Thank you for pointing out this issue. We have verified the full names and abbreviations you mentioned one by one and made the following improvements. For the term *'KL divergence'* you pointed out, we have provided its full form as *'Kullback-Leibler divergence'* when it first appeared to ensure the professionalism and accuracy. In addition, we conducted a thorough check of other related terms and discovered that the abbreviation *'MINE'* was also not expanded the first time it appeared. Its full name is *'Mutual Information Neural Estimation'* and we have added this clarification in the manuscript. We will ensure that all key terms in the manuscript provide their full names when they first appear and are clearly presented in the final version.
>
> **Q2: It is seen that the improvement of the proposed method over other methods is significant. Will the source code be released to the public to enhance the development of the multi-modal clustering community？**
> ***Response：*** We sincerely appreciate your recognition of the effectiveness of the proposed SDCIB. We fully understand the importance of code availability in promoting research transparency and advancing the field of multi-modal clustering. Currently, we have organized and optimized the code to ensure its clarity and readability. We plan to release the source code after the official publication of the paper, hoping that it will contribute to the research community and support further advancements in this field.
>
> **Q3: Are there any limitations of the proposed method? The authors are encouraged to give them in the conclusion.**
> ***Response：*** We sincerely appreciate your valuable suggestion. In response, we have revised the conclusion to include the following limitations of the proposed SDCIB:
> * The proposed SDCIB shows limited performance when handling incomplete data samples, particularly when certain modalities or features are missing. In such cases, the model may struggle to accurately learn the relationships between the modalities, which could lead to suboptimal clustering results.
> * The number of clusters must be known in advance as most existing multi-modal clustering methods. This requirement can be restrictive, as it assumes prior knowledge of the data's underlying structure.
> * The proposed SDCIB primarily relies on batch learning, which may not be suitable for certain applications, such as streamed multi-modal data.
>
> Additionally, we acknowledge that while these limitations may affect certain scenarios, we will explore solutions to address them, with the aim of further enhancing the method’s robustness and applicability.
>
> Thanks again for the valuable suggestions provided by the reviewer. The modifications will be added to the final version.

---

### Official Review · Reviewer_t5hp · 2025-03-10

**Overall Recommendation:** 3

**Summary:**

In multi-modal clustering, effectively capturing the complex relationships between modalities remains a challenge. For solving this, this paper propose a new super deep contrastive information bottleneck method to maximize the utilization of latent information in multi-modal data. It firsts introduces hidden layer information from the encoder into the clustering process to enhance modality feature representation; then, it proposes a dual contrastive learning optimization strategy The experimental results demonstrate that the method not only enhances clustering performance but also exhibits strong applicability in multi-modal data modeling.

**Claims And Evidence:**

yes

**Essential References Not Discussed:**

No essential related works are missing in the paper.

**Experimental Designs Or Analyses:**

Yes, I have checked. By conducting experiments on four multi-modal datasets, the method significantly outperforms existing methods.

**Methods And Evaluation Criteria:**

yes

**Other Comments Or Suggestions:**

I have no other comments and suggestions.

**Other Strengths And Weaknesses:**

The proposed method is based on an information-theoretical method called information bottleneck which is named SDCIB and is also organized in a rigorous, theoretical sounded way. The descriptions is clear and well written, and is novel enough for this conference. I have my comments below:

1) It is good to see that the authors give some recent IB works on MMC problem, it is suggested to give a deeper analysis on the limitations of them. Although the differences with the proposed method is given now, some more analysis is also needed.

2) Some equation references is not proper, such as Eq. 2, Eq. 8. A bracket is missing throughout the whole manuscript.

3) Some English usage about the writing details is suggested to be improved, such as 'cluster number K'.

**Questions For Authors:**

I have no other questions for this paper.

**Relation To Broader Scientific Literature:**

This paper propose a new super deep contrastive information bottleneck method to solve the multi-modal clustering problem.

**Theoretical Claims:**

No theoretical claims here.

---

> ### Author Rebuttal · Authors · 2025-03-31
>
> Thank you for the insightful comments and constructive suggestions. We have carefully revised the whole manuscript and provided detailed responses to each point below.
>
> **Q1: It is good to see that the authors give some recent IB works on MMC problem, it is suggested to give a deeper analysis on the limitations of them. Although the differences with the proposed method is given now, some more analysis is also needed.**
> ***Response：*** Thank you for the insightful comments on the information bottleneck work in this paper.
> Based on your suggestions, we conducted a more in-depth analysis of the limitations of recent IB works on MMC problems. These analyses were added to Section 2.1 of the manuscript as follows:
> Federici et al. [1] proposed a multi-modal IB method that can identify non-shared information between two modalities, but it only explores the correlation of different modalities through feature distribution, ignoring the consistency of cluster assignment, making the learned feature representation unfriendly to downstream clustering tasks. Yan et al. [2] proposed a multi-modal IB method that uses shared representations of multiple modalities to eliminate private information of a single modality. Yan et al. [3] further proposed an incremental IB method that builds acknowledge base to solve the clustering problem of incremental modalities. Both of the above works considered the consistency of feature distribution and cluster assignment at the same time, but they failed to consider the correlation between feature distribution and clustering results. All the above MMC IB methods ignore the rich information contained in the hidden layers of the encoder and fail to explicitly utilize it.
> The above limitations motivate us to the proposed SDCIB.
> **References**
> [1]: Federici, M., Dutta, A., Forré, P., Kushman, N., and Akata, Z. Learning robust representations via multi-view information bottleneck. arXiv preprint arXiv:2002.07017, 2020.
> [2]: Yan, X., Mao, Y., Ye, Y., and Yu, H. Cross-modal clustering with deep correlated information bottleneck method. IEEE Transactions on Neural Networks and Learning Systems, 2023.
> [3]: Yan, X., Mao, Y., Ye, Y., and Yu, H. Incremental multiview clustering with continual information bottleneck method. IEEE Transactions on Systems, Man, and Cybernetics: Systems, 2024.
>
> **Q2: Some equation references is not proper, such as Eq. 2, Eq. 8. A bracket is missing throughout the whole manuscript.**
> ***Response：*** Thank you for the detailed review. In response to your specific suggestion regarding the equation references format, we have made thorough revisions to the whole manuscript. First, based on your feedback, we corrected Eq. 2 and Eq. 8 to the standardized format, Eq. (2) and Eq. (8), respectively. Additionally, we conducted a comprehensive review of the whole manuscript to ensure consistency and accuracy in all equation references. Furthermore, in order to further improve the quality of the manuscript, we not only addressed the equation references issue but also meticulously checked and adjusted all formatting details throughout the whole manuscript (including formulas, symbols, references, etc.) to ensure greater precision and rigor.
>
> **Q3: Some English usage about the writing details is suggested to be improved, such as 'cluster number K'.**
> ***Response：*** Thank you for the attention to the language details in our manuscript and for the helpful suggestions. We have carefully reviewed the whole manuscript for language issues and made detailed corrections where needed. As you suggested with *'cluster number $K$'*, we have corrected it to *'the number of clusters $K$'*, making it more standard. Additionally, we have checked for similar language issues throughout the manuscript and have made the necessary adjustments, such as *'the parameter $α$, and the parameter $β$'*, which has now been adjusted to a more fluent *'the parameter $α, β$'* to improve the accuracy and naturalness of the expression. To ensure the overall language quality, we have conducted an in-depth review of the whole manuscript's English expressions, optimizing wording and grammar to make the overall presentation more standardized and clear.
>
> Thanks again for the valuable suggestions provided by the reviewer. The modifications will be added to the final version.

---

### Decision · Program_Chairs · 2025-05-01

**Decision:**

Accept (poster)

**Comment:**

This paper focuses on how to deeply explore the complex latent information and interdependencies between modalities in multi-modal clustering. The authors introduce the rich information contained in the encoder's hidden layers into the loss function to mine both modal features and the hidden modal  relationships. They perform dual optimization by concurrently considering consistency information from both the feature distribution and clustering assignment perspectives to improve the clustering accuracy and robustness.  Experimental results illustrate the effectiveness of the presented method.

After rebuttal, the raised concerns are solved, all reviewers are positive about this manuscript and recognize its contributions.  After reading the manuscript and responses, I agree with reviewers' opinions and recommend an acceptance for this manuscript.